

# Exact one-particle density matrix for SU(N) fermionic matter-waves in the strong repulsive limit

Andreas Osterloh[1], Juan Polo[1], Wayne J. Chetcuti[1,2,3] and Luigi Amico[1,3,4⋆]

**1** Quantum Research Center, Technology Innovation Institute, Abu Dhabi, P.O. Box 9639, UAE
**2** Dipartimento di Fisica e Astronomia, Via S. Sofia 64, 95127 Catania, Italy
**3** INFN-Sezione di Catania, Via S. Sofia 64, 95127 Catania, Italy
**4** Centre for Quantum Technologies, National University of Singapore, 3 Science Drive 2, Singapore 117543, Singapore

## Abstract

We consider a gas of repulsive $N$-component fermions confined in a ring-shaped potential, subjected to an effective magnetic field. For large repulsion strengths, we work out a Bethe ansatz scheme to compute the two-point correlation matrix and then the one-particle density matrix. Our results hold in the mesoscopic regime of finite but sufficiently large number of particles and system size that are not accessible by numerics. We access the momentum distribution of the system and analyse its specific dependence of interaction, magnetic field and number of components $N$. In the context of cold atoms, the exact computation of the correlation matrix to determine the interference patterns that are produced by releasing cold atoms from ring traps is carried out.

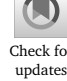

## Contents

⋆ On leave from the Dipartimento di Fisica e Astronomia "Ettore Majorana", University of Catania.

# 1 Introduction

In low-dimensional many-body systems, quantum fluctuations are particularly pronounced, and therefore even a weak interaction can lead to dramatic correlations. Such a simple fact makes the physics of $1d$ many-body systems exotic and distinct from the physics of higher dimensional systems [1]. The breakdown of the Fermi liquid paradigm and Luttinger liquid behaviour, the spin-charge separation in fermionic systems, elementary excitations with fractional statistics and Haldane order are just some of the characteristic traits addressed in the last few decades of research on the subject [2–5]. One dimensional systems can be realized by confining the spatial degrees of freedom, as in quantum wires [6], in chains of Josephson junctions [7] or in certain classes of polymers [8]; in other instances, the dimensionality is constrained dynamically, as in carbon nano-tubes [9], edge states in quantum Hall effect [10] or in metals with dilute magnetic impurities [11]. With the advent of quantum technology, seeking quantum correlations as a resource, the impact of $1d$ physics has been considerably widened. In this paper, we will be dealing with strongly correlated $N$-component fermions confined in one spatial dimension. The two-component electronic case is ubiquitous in physical science from condensed matter to high energy physics and clearly relevant for a large number of technological applications. Systems with $N > 2$ have emerged as effective descriptions in specific condensed matter or mesoscopic physics contexts [12–15].

Recently, the relevance of N-component fermions has been significantly boosted through the experimental realizations of alkaline earth-like fermionic atomic gases [16–19]; in there, the two-body interactions resulted to be SU($N$)-symmetric, reflecting the absence of hyperfine coupling between the atoms' electronic and nuclear degrees of freedom [20–22]. Such artificial matter is relevant for high precision measurements [23, 24] and has the potential of considerably expanding the scope of cold atoms quantum simulators [16, 25–28]. Here, we focus on SU($N$) fermions described by a Hubbard type model [20, 22]. In the dilute regime of less than one particle per site, the lattice model captures the physics of continuous systems with delta-interaction [29], which is exactly solvable by Bethe Ansatz [30, 31]. Exact solutions of $1d$ interacting quantum many-body systems play a particularly important role since their physics is often non-perturbative, with properties that are beyond the results obtained with approximations [1]. As such, exact results, though rare and technically difficult to achieve, form a precious compass to get oriented in the $1d$ physics.

Here, we provide the exact expression of the two-point correlation matrix of fermions with $N$ components, determining the one-body density matrix, in the limit of strong particle-particle interactions. We consider particles confined in a ring-shaped potential subjected to an external magnetic flux in the limit of large repulsive interactions. We work in the mesoscopic regime in which such a magnetic field is able to start an $N$-component fermionic matter-wave persistent current. We analyze the distribution of the momentum of particles, which, despite being one

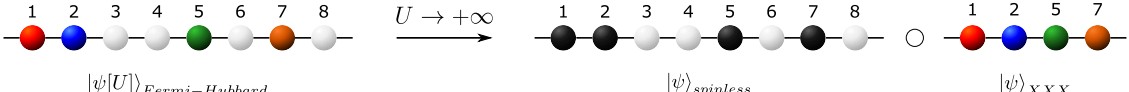

Figure 1: Schematic representation of the decoupling of the SU($N$) Fermi-Hubbard into the spinless and Heisenberg-$XXX$ Hamiltonians at infinite repulsive interactions $U$. On the left, the figure depicts the SU(4) Hamiltonian with one-particle per colour and 4 empty sites (white). On the right, we have the spinless Hamiltonian with 4 fermions (black) and 4 empty sites. In addition, there is the SU(4) Heisenberg Hamiltonian with one spin in each orientation. Note that after the decoupling, the index corresponding to a given colour in the Heisenberg Hamiltonian changes in order to accommodate the new framework, but the arrangement of the colours in the original chain is maintained. The circle indicates a function composition in a mathematical sense: $f \circ g = f(g)$.

of the simplest correlations, is able to reflect certain effects of the interaction [32]. On the technical side, we point out that, despite its simple expression, the momentum distribution can only be calculated numerically for a small number of particles and is even less accessible when considering the strongly correlated regimes. Even for integrable models, it is not manageable, especially in the mesoscopic regime of finite but sufficiently large particle systems. The case $N = 2$ in the absence of magnetic flux was discussed by Ogata and Shiba [33].

The one-body density matrix plays a crucial role in different schemes of time-of-flight expansions in cold atoms settings [28, 34–38]. The effect of an artificial magnetic field in neutral two-component fermions confined in tight toroidal-shaped potentials was explored in recent experiments [39, 40]. The arising persistent current pattern is produced as a result of specific transitions between suitable current states characterized by different particles' spin configuration [41, 42]. We will show how to handle the extra-complications coming from the above ground-states transitions in computing the correlation matrix of the system for different magnetic fluxes.

The paper is structured as follows. In Sec. 2 we discuss the model describing our system and introduce the spin-charge decoupling mechanism. In Sec. 3 and Sec. 4 we present the results achieved for the momentum distribution and interference dynamics of SU($N$) fermions. Conclusion and outlooks are given in Sec. 5.

## 2  Model and methods

The one-dimensional Hubbard model for $N$-component fermions residing on a ring-shaped lattice comprised of $L$ sites, threaded by an effective magnetic flux $\phi$ reads

$$\mathcal{H} = -t \sum_{j}^{L} \sum_{\alpha=1}^{N} (e^{i\frac{2\pi\phi}{L}} c_{j,\alpha}^{\dagger} c_{j+1,\alpha} + \text{h.c.}) + U \sum_{j}^{L} \sum_{\alpha<\beta} n_{j,\alpha} n_{j,\beta} \,, \tag{1}$$

where $c_{j,\alpha}^{\dagger}$ creates a particle with colour $\alpha$ on site $j$ and $n_j = c_j^{\dagger} c_j$ is the local particle number operator. $U$ and $t$ denote the interaction and hopping strengths respectively. In this paper, we consider only the repulsive case such that $U > 0$. The Peierls substitution $t \to t e^{i\frac{2\pi\phi}{L}}$ accounts for the gauge field. In standard implementations such a field can be an actual magnetic field, while it can be artificially created in cold atom settings [43].

For $N = 2$, the model in Eq. (1) is Bethe ansatz solvable for all system parameters and filling fractions $\nu = N_p/L$ [44]. For $N > 2$, Bethe ansatz solvability holds for the continuous

limit of vanishing lattice spacing, with the model turning into the Gaudin-Yang-Sutherland model, that describes SU($N$) symmetric fermions with delta interactions [22, 30, 31]. This limit is achieved when considering the dilute regime, such that $\nu \ll 1$ [29]. In the following, we will refer to the Bethe ansatz solution of the SU($N$) Hubbard model in this limit.

Accordingly, within a given particle ordering $x_{Q_1} \leq \ldots \leq x_{Q_{N_p}}$, the eigenstates of the model (1) can be expressed as

$$f(x_1, ..., x_{N_p}; \alpha_1, ..., \alpha_{N_p}) = \sum_P A(Q|P) \varphi_P(\alpha_{Q1}, ..., \alpha_{QN_p}) \exp\left( i \sum_{j=1}^{N_p} k_{Pj} x_{Qj} \right), \qquad (2)$$

where $A(Q|P) = \mathrm{sign}(P)\mathrm{sign}(Q)$ with $P$ and $Q$ being permutations introduced to account for the eigenstates' dependence on the relative ordering of the particle coordinates $x_j$ and quasi-momenta $k_j$, with $\varphi$ being the spin wavefunction. The latter accounts for all different components of the system, which can be obtained by nesting the Bethe ansatz [31]. As a result, the spin-like rapidities for each additional colour $\Lambda_\alpha$, which are the conserved quantities for the SU($N$) degrees of freedom (see Appendix A.1), are all housed in $\varphi$ [31, 45]. In particular, we note that the ground-state of the system correspond to real $k_j, \Lambda_\alpha$.

Despite the access to the energy spectrum is greatly simplified due to integrability, the calculation of the exact correlation functions remains a very challenging problem [46], especially in the mesoscopic regime of large but finite $N_p$ and $L$ [47].

Here, we will be focusing on the large $U$ limit where the correlation functions become addressable as we shall see. The simplification arises because the charge and spin degrees of freedom decouple (such a decoupling occurs only for states with real $k_j$) [33, 41, 42]. The decoupling is manifested in the Bethe equations of the system. In the limit $U \to \infty$, the charge degrees of freedom are specified as (see Appendix A.1.1 for a sketch of the derivation):

$$k_j = \frac{2\pi}{L}\left[ I_j + \frac{X}{N_p} + \phi \right], \qquad (3)$$

where $I_j$ are the charge quantum numbers of the spinless fermionic model and $X = \sum_{\ell}^{N-1} \sum_{\beta_\ell}^{M_\ell} J_{\beta_\ell}$ denotes the sum of the spin quantum numbers. As an effect of the spin-charge decoupling, each wavefunction amplitude can be written as a product between a Slater determinant of spinless fermions $\det[\exp(ik_j x_{Qj})]$ and a spin wavefunction $\varphi(y_1, \ldots, y_M)$ [33]

$$f(x_1, ..., x_{N_p}; \alpha_1, ..., \alpha_{N_p}) = \mathrm{sign}(Q)\det[\exp(ik_j x_l)]_{jl}\varphi(y_1, \ldots, y_M). \qquad (4)$$

Consequently, in the limit of $U \to +\infty$ these states of the Hubbard model can be written as

$$|\psi[U]\rangle_{\text{Fermi–Hubbard}} \overset{U \to +\infty}{\longrightarrow} |\psi[\psi_{XXX}]\rangle_{\text{spinless}}. \qquad (5)$$

The logic of the decoupling occurring in the wavefunction is depicted in Fig. (1). It is important to emphasize that this is not a tensor product but corresponds to a composition of functions.

The XXX Heisenberg model in Eq. (5) is also integrable for SU($N$) and all $1 < N \in \mathbb{N}$. The corresponding Hamiltonian can be constructed as a sum of permutation operators $\mathcal{H}_{XXX} = \sum_i P_{i,i+1}$ [22, 48], where $P_{i,i+1}$ can be expressed in terms of SU($N$)-generators. The Hamiltonian $P_{i,i+1}$ permutes SU($N$) states on sites $i$ and $i+1$ (see also Appendix (A.2) and Eq. (A.22)). Even though Bethe ansatz integrable, the exact access of explicit expression of the eigenstates of the antiferromagnet Heisenberg model is very challenging. In our paper, therefore, the quantum state is obtained by combining the Bethe ansatz analysis with the Lanczos numerical method. The procedure is described below.

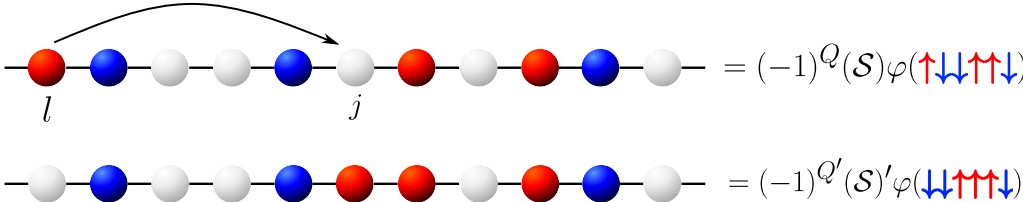

Figure 2: Schematic representation of the effect of $c^{\dagger}_{l,\alpha} c_{j,\alpha}$ on an SU(2) wavefunction. The upper part depicts the initial state in a given configuration with the corresponding decoupled wavefunction shown on the right. The bottom figure illustrates the final state and its corresponding wavefunction after performing the hopping action on the initial state. This figure is adapted from [33].

**Finding the ground-state.** Firstly, we note that for each non-degenerate ground-state of the Hubbard model, there exists a corresponding single eigenstate of the Heisenberg model. In principle, such a state-to-state correspondence could be obtained by identifying the spin quantum numbers labeling the states of the Hubbard model (through the Bethe ansatz equations) with the quantum numbers for the Heisenberg model. However, as mentioned above such a procedure is quite involved when trying to access to the quantum states. Therefore, we use a combination of Bethe ansatz and numerical methods: i) inserting the spin quantum numbers characterizing a given state in the Hubbard model into the Heisenberg Bethe ansatz, enables us to calculate the correct energy, which is then matched with the numerically obtained spectrum of the anti-ferromagnet; then ii) the SU($N$) quadratic Casimir operators (see the Appendix A.2.2) are used to characterize the total SU($N$)-spin of the states. The Casimir operators are commuting with the whole SU($N$) group and hence are constants of the motion of both the Heisenberg Hamiltonian and the SU($N$) Hubbard model. In particular, we note that the Casimir operator for $N = 2$ corresponds to the total spin operator squared $\vec{S}^2$. For $\phi = 0$, the state of the Hubbard model is non-degenerate. Therefore, this approach can uniquely characterize the states. For $\phi \neq 0$ however, it results that the energy of the Heisenberg model is degenerate as is the Casimir value. This degeneracy can be resolved for the SU(2) case by looking at the permutation operators $P_{j,j-1}$ (such operators do not commute with the Heisenberg Hamiltonian by construction). For larger $N$ and $\phi \neq 0$ we do not have a general method. However, we note that degenerate states with the same Casimir value consist of different projections into the Heisenberg basis, which allow us to uniquely identify the correct ground-states to be taken at increasing flux [42] (see the Appendix for a detailed explanation).

We found that non-degenerate ground-states with odd and even number of particles per species correspond to different values of the Casimir operators, and therefore to different representations of the SU($N$) algebra.[1] The corresponding states are hence chosen based on the parity of the species occupation number.

We comment that, for $N_p = (2m)N$ fermions with integer $m$ at zero flux, the ground-state wavefunction of the Hubbard model is not a singlet in contrast with that of the anti-ferromagnetic Heisenberg model. In the case of SU(2), this issue was circumvented by considering anti-periodic boundary conditions for the Hubbard model, which results to be a singlet ground-state [33]. In contrast with the method presented in [33], we do not modify the boundary conditions for model (1) but instead we modify the spin quantum numbers in Eq. (3) such that the non-degenerate triplet eigenstate of the Heisenberg model is selected.

Our proposed scheme is reliant on model (1) being integrable. As stated beforehand, one instance of integrability occurs for dilute filling fractions, such that the model turns into the

---

[1] It is worth noticing that this eigenvalue may be accidentally degenerate in the Heisenberg model.

Gaudin-Yang-Sutherland model. In what follows, the system sizes considered are far from being in the dilute limit. Nonetheless, we find that our method is still applicable in this regime (see section A.4 in the appendix), since in the limit of infinite repulsion, the probability of having more than two particles interacting is vanishing, thereby satisfying the Yang-Baxter condition for integrability [49]. Indeed, for the low-lying spectrum and the corresponding correlations, such a statement was verified by comparing with exact diagonalization (see Table 1 in the Appendix). It is worth remarking that the numbers of $N_p$ and $L$ considered in this paper would correspond to a Hilbert space size, that is intractable with exact diagonalization. On account of the spin-charge decoupling, we are able to separate the problem into the spinless and Heisenberg parts, resulting in smaller Hilbert spaces, making systems with large values of the parameters accessible (see Appendix A.4).

**The one-body density matrix.** In the present work, we apply the factorization (5) to determine the one-particle density operator through the calculation of the two-point correlation matrix of the SU($N$) Hubbard model (1), together with its dependence with the flux $\phi$:

$$\langle \Psi_\alpha(x)^\dagger \Psi_\alpha(x) \rangle = \sum_{l,j} w^*(x-x_l) w(x-x_j) \langle c^\dagger_{l,\alpha} c_{j,\alpha} \rangle, \tag{6}$$

where $\Psi_\alpha^\dagger(x)$ and $\Psi_\alpha(x)$ are fermionic field operators satisfying $\{\Psi_\alpha^\dagger(x), \Psi_{\alpha'}(y)\} = \delta(x-y)\delta_{\alpha,\alpha'}$. The above equation is obtained by expanding the field operators into the basis set of single band Wannier functions $w(x)$ (that we take to be independent of the specific $N$ component) such that $\Psi(x) = \sum_j^L w(x-x_j) c_j$.

The spin-charge decoupling is attained through the Bethe equations. Subsequently, the spectrum of the Heisenberg model is obtained through exact diagonalization. In line with methodology outlined in the previous section, we point out that one can make use of DMRG [50,51] to certify that the chosen state from the Heisenberg spectrum has the same total spin as its Hubbard counterpart. Even though DMRG is known to have issues in the limit of large interaction and large degree of state degeneracies, it can still be utilized for intermediate interactions.

The energy scale is fixed by $t=1$ and only systems with an equal number of particles per component are considered.

## 3 Momentum distribution

The momentum distribution is defined as

$$n_\alpha(k) = \frac{1}{L} \sum_{j,l}^L e^{ik(r_l-r_j)} \langle c^\dagger_{l,\alpha} c_{j,\alpha} \rangle, \tag{7}$$

with $r_j$ denoting the position of the lattice sites in the ring's plane and is normalized to the occupation number of each species. In the aforementioned limit of infinite repulsion, the correlation matrix can be recast as

$$\langle c^\dagger_{l,\alpha} c_{j,\alpha} \rangle = \sum_{\{\text{config.}\}} \text{sign}(Q)\text{sign}(Q')(\mathcal{S})^*(\mathcal{S})'\omega(j \to l, \alpha), \tag{8}$$

where $\mathcal{S}$ denotes the Slater determinant of the charge degrees of freedom, and $Q$ refers to the sign of the corresponding permutation. $\mathcal{S}$' and $Q$' are the same quantities but evaluated for the wavefunction of a fermion that moved from the $j$-th to the $l$-th site (see Fig. 2). We note that one has to account for the shift in the quasi-momenta $k_j$ induced by the spin quantum numbers

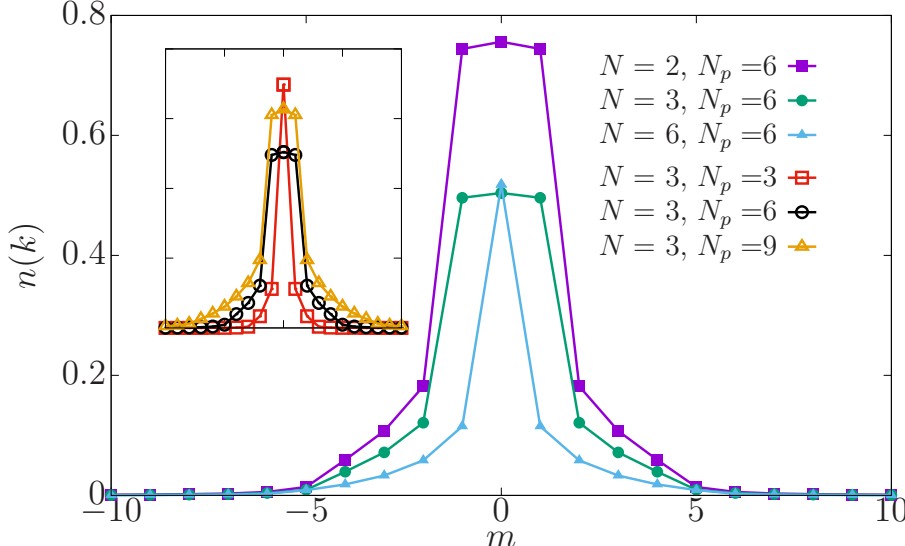

Figure 3: The momentum distribution for different SU($N$) and ratios $N_p/N$. Main panel shows the momentum distribution, normalized to the occupation number of each species, for a fixed number of particles $N_p = 6$, while the insets displays the momentum distribution for a fixed $N = 3$ but different number of particles. In both panels we showcase the interplay between occupation and SU($N$) character of the system. The system size is fixed to $L = 27$, with the integers $m$ corresponding to the momenta $2\pi m/L$.

through Eq. (3). These quasimomenta are different from the momenta $k$ of the lattice in the momentum distribution discussed here. Furthermore, we would like to emphasize that instead of calculating the Slater determinant for the continuous Gaudin-Yang-Sutherland model, we discretize it. Such an approach is necessary in order to keep track of the mapping between the spin wavefunctions of the Hubbard and Heisenberg models. This justification is numerically supported in Table 1 in the Appendix. The term $\omega(j \to l, \alpha)$ corresponds to the spin part of the wavefunction of the Hubbard model, taking into account the sum over all the spin configurations and any changes in $\varphi(y_1, \ldots, y_M)$.

Before proceeding to evaluate Eq. (8), we note that $\omega(j \to l, \alpha)$ is independent of $\alpha$: $\omega(j \to l, \alpha) = \omega(j \to l)$. Moreover, in the limit of infinite repulsion, the spin wavefunction of the Hubbard model corresponds and can be mapped to that of the Heisenberg such that $\omega(j \to l) = \tilde{\omega}(j' \to l')$, where the tilde indicates the spin correlation function of the Heisenberg model. In this mapping, we associate the $j'$th spin of the Heisenberg model to the fermion on the $j$th site of the Hubbard model, that after the hopping operation $c_l^\dagger c_j$ becomes the $l'$th spin corresponding to an electron of the $l$th site – see Fig. (2). We emphasize that the expression in Eq. (8) is of the same form as for the SU(2) case [33]. The difference lies in the definition of $\tilde{\omega}(j, l)$, which encodes the SU($N$) character of the system:

$$\tilde{\omega}(j' \to l') \equiv \langle P_{l', l'-1} P_{l'-1, l'-2} \cdots P_{j'+1, j'} \rangle_H. \tag{9}$$

This corresponds to the expectation value in the Heisenberg state of the SU($N$) permutation operator $P_{j', j'-1}$ that exchanges the $j'$th and $(j'-1)$th sites.

With the states obtained as summarized above, we evaluate the momentum distribution $n(k)$ in Eq. (7). In Fig. 3, the momentum distribution in the absence of magnetic flux is presented for different SU($N$). For a fixed $N_p$ and increasing $N$, the momentum distribution is observed to be less broad and to be more centralized around $k = 0$. This is to be expected since as $N \to \infty$, SU($N$) fermions emulate bosons in terms of level occupations [38, 52].

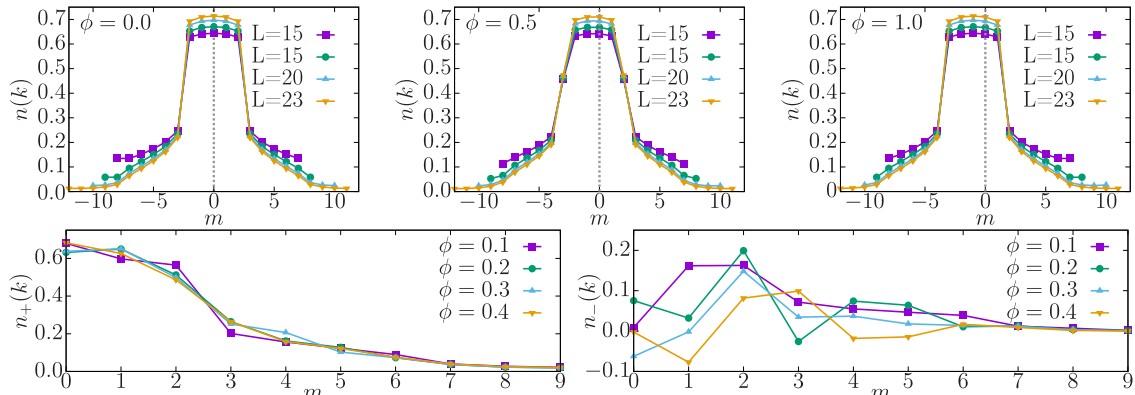

Figure 4: Above: The momentum distribution function $n(k)$ for 10 particles and various sites for SU(2) fermions. One can see that the flux essentially shifts the momentum distribution as expected. Below: We plot the symmetric (left) and anti-symmetric (right) components of the momentum distribution denoted as $n_+(k) = [n(k) + n(-k - \Delta k)]/2$ and $n_-(k) = n(k) - n(-k - \Delta k)$ respectively as a function of the effective magnetic flux $\phi$. We note that these intermediate values of the flux produces a momentum distribution that is non-symmetric. The integers $m$ correspond to the momenta $2\pi m/L$.

Conversely, for fixed $N$ and increasing $N_p$, the momentum distribution reflects the fermionic statistics of the system, as it becomes broader due to the occupation of different momenta (see inset of Fig. 3).

Fig. 4 depicts the momentum distribution for an SU(2) symmetric system in the presence of an effective flux. In this case, the ground-state of the Hubbard model is characterized by level crossings to counteract the flux imparted to the system [41,42]. Such level crossings correspond to different Heisenberg states, which can be obtained with the previously mentioned procedure in Sec. 2 by an appropriate change in spin quantum numbers (see Appendix A.3.1). From the top row of Fig. 4, it is clear that the effect of the magnetic flux manifests itself as a shift in the momentum distribution: shift gets progressively larger with increasing flux. To capture how this happens precisely in the momentum distribution, we plot the symmetric and anti-symmetric components of the momentum distribution denoted as $n_+$ and $n_-$ respectively in the bottom panel of Fig. 4.

### 3.1 The Fermi gap for $U = \infty$

In the thermodynamic limit at temperature $T = 0$ and $U = 0$, the Fermi function drops from a finite value to zero at the Fermi momentum $k_f$. At finite $U$, states with $k > k_f$ can be occupied and, compared with the $U = 0$ case, the gap at $k_f$ is reduced accordingly. The Fermi gap is known as the quasi-particle weight in higher dimensions and is related to the poles of the Green function with positive imaginary parts [53–55]. For SU($N$) symmetric particles the maximum occupation of a single $k$-level is $N$. Consequently, the Fermi-distribution for $N \to \infty$ should become a Bose-distribution (which has no gap). Therefore, this Fermi gap $\Delta$ must tend to zero in this limit.

Since we consider finite number of particles and system size, our system is far from the thermodynamic limit. We note that parity effects appear in $N_p/N$ for SU($N$) fermions [42]. Therefore we distinguish the two cases: odd occupations ($N_p/N$ odd) and even occupations ($N_p/N$ even). Defining the gap for the odd occupations is straight forward: every single $k$-level up to $|k_f|$ is occupied for $U = 0$. For example, in the case of SU(2), $N_p = 6$, all $k \in \{0, 1, -1\}$ are fully occupied. Therefore, the gap is defined as $f(k_f) - f(k_f + \Delta k)$, with $\Delta k = 2\pi/L$,

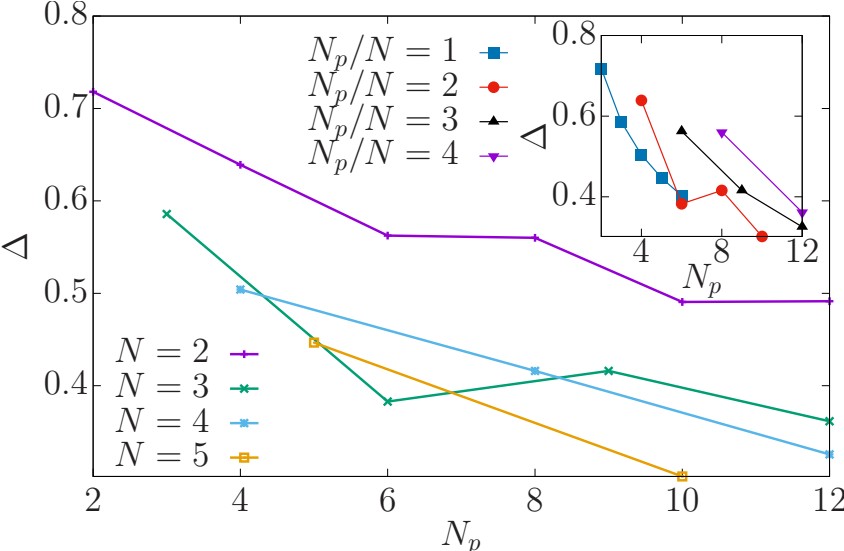

Figure 5: Gap in the Fermi-distribution $\Delta$ as function of $N_p$ in the limit of infinite repulsion. It is shown for different cases for dependencies of $N$ in SU($N$) (left) and for different values of $N_p/N$ from 1 to 4 for $L = 27$. It can be seen that the gap decreases with growing $N$. The single exception is the case of SU(3) (see left panel). The particularities concerning definition of the gap for finite number of particles is described are the main text.

where $f(k)$ corresponds to the Fermi-distribution function. The situation is different for an even occupation per species. In this case, the levels $|k_f|$ are only partially filled and this is visible even for $U = 0$ and finite number of particles, where a single level appears within the gap. However, this single momentum state does not enter the definition of the Fermi gap. Therefore, we define the gap in this case as $f(k_f - \Delta k) - f(k_f + \Delta k)$. Note that in one dimension the Fermi-distribution function in the thermodynamic limit becomes a weak singularity for the Luttinger liquid [32]. In our case, we cannot distinguish a gap from a weak singularity as we are far away from the thermodynamic limit.

In agreement with the above argument, we find that the gap is generically going down with $N$, but with a non-trivial dependence on $N_p$ and with parity effects for SU(2) and more pronounced for SU(3) - see Fig. 5. Specifically, we note that the two particles per species in the SU(3) symmetric case has a non-monotonous behavior with respect to the trend of decreasing gap with growing $N_p$, that is present for the other curves. This behavior might be attributed to parity effects but larger systems, not attainable with current techniques, would be needed to investigate such behavior. Additionally, grouping the different gaps as a function of $N_p/N$ does not lead to strictly decreasing behavior (see inset of Fig. 5). However, we note that expecting the gap to decrease for $N_p$ fixed with growing $N$, would give a hint towards a parity effect of the number of components $N$, at least in the case $N_p = 6$. In principle, the Fermi gap need not follow a monotonic behavior. The expectation is that for each $N_p$ it has to eventually converge to zero as $N \to \infty$, corresponding to bosonic behaviour as mentioned previously. Lastly, it is important to notice that in the systems considered in this paper, we never come below the ratio of $N_p/N = 1$ because we fixed the occupation of each component being the same.

# 4 Interference dynamics in ultracold atoms

In this section, we present a particular scenario in which the exact one-body density matrix can be tested in current state-of-the-art experimental observables in ultracold atom settings. Specifically, we consider homodyne [39] and self-heterodyne [40] protocols following the recent experiments carried out in fermionic rings.

The homodyne protocol consists in performing time-of-flight (TOF) imaging of the spatial density distribution of the atomic cloud: upon sudden release from its confinement potential, the atomic cloud expands freely, with the initially trapped atoms interfering with each other creating specific interference patterns. The resulting inteference pattern depends on the correlations that the particles have at the moment in which atomes are released. The TOF image can be calculated as

$$n_\alpha^{(TOF)}(\mathbf{k}) = |w(\mathbf{k})|^2 \sum_{j,l}^{L} e^{i\mathbf{k}(\mathbf{r}_l - \mathbf{r}_j)} \langle c_{l,\alpha}^\dagger c_{j,\alpha} \rangle , \tag{10}$$

where $w(\mathbf{k})$ is the Fourier transform of the Wannier function, $\mathbf{r}_j$ denotes the position of the lattice sites in the ring in the plane and $\mathbf{k} = (k_x, k_y)$ are their corresponding Fourier momenta. Note that we have taken the zeroth order of $w(x)$ through the harmonic approximation.

The self-heterodyne protocol follows the same procedure as the homodyne one, albeit with an additional condensate placed in the center of the system of interest, to act as a phase reference. Accordingly, as the center and the ring undergo free co-expansion in TOF, characteristic spirals emerge as the two systems interfere with each other and current is present in the system. In order to observe the phase patterns in a second quantized setting, one needs to consider density-density correlators between the center and the ring [37, 56]: $G_{R,C} = \sum_\alpha \sum_{j,l} I_{jl}(\mathbf{r}, \mathbf{r}', t) \langle c_{l,\alpha}^\dagger c_{j,\alpha} \rangle$, where $I_{jl}(\mathbf{r}, \mathbf{r}', t) = w_c(\mathbf{r}', t) w_c^*(\mathbf{r}, t) w_l^*(\mathbf{r}' - \mathbf{r}_l', t) w_j(\mathbf{r} - \mathbf{r}_j, t)$. By exploiting the correlation matrix we calculated in the previous sections, we obtain the interference images that are obtained through the two above sketched expansion protocols for two-component fermions exactly - see Fig. 6. The left panel displays a cut on the TOF momentum distribution (at $k_y = 0$), with the inset depicting the same quantity in the $k_x$-$k_y$ plane. The right panels of Fig. 6 show the self-heterodyne interference pattern at zero and half flux quantum. These display the characteristic dislocations (radially segmenting lines) that at strong interactions were shown to depend on particle number, number of components and flux [37]. On going to the limit of infinite repulsion, the energy and consequently the persistent current landscape, changes from being periodic with the bare flux quantum $\phi_0$ to displaying a reduced periodicity of $\phi_0/N_p$ irrespective of the SU($N$) symmetry of the system (see Fig. 7 in the Appendix). As such, the ground-state energy goes from being a single parabola at zero interaction, to having $N_p$ parabolic segments, that in the Bethe ansatz language are characterized by different spin quantum numbers. Remarkably, these different parabolas manifest as a result of different energy level crossings to counteract the flux threading the system, resulting in an effective fractionalization of the current. In [37] it was discussed how this fractionalization can not only be monitored but also visualized in self-heterodyne interferograms, which exhibits a different number and orientation of the dislocations for the different parabolas. In the left panel of Fig. 6, we see that such dislocations are captured by our proposed scheme giving us access to the infinite repulsive limit in an exact way.

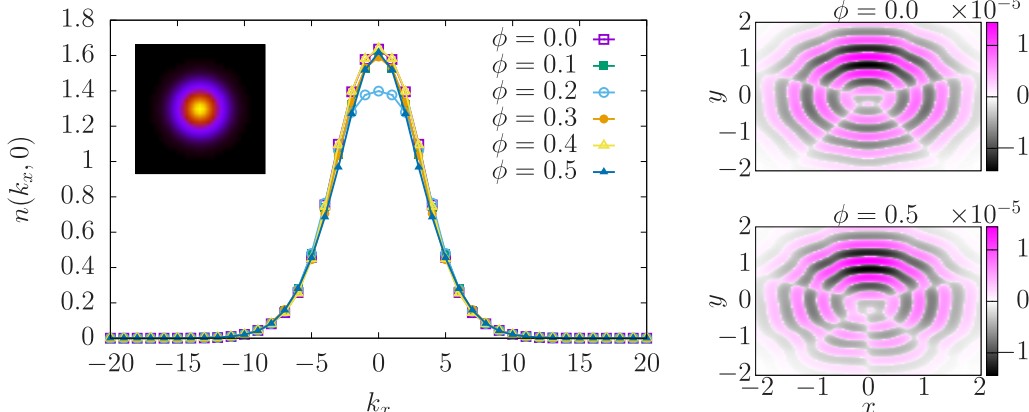

Figure 6: Interference patterns for SU(2) $N_p = 10$ particles residing in $L = 15$ sites. Left panel shows a cut of the TOF momentum distribution for different fluxes, the last one (at half flux quantization) displaying a reduction in its maximum value. Inset displays the full TOF for at zero flux quantum. Right panels display the self-heterodyne interference at zero and half flux quantum. Both show the characteristic interference pattern and dislocations found in strongly interacting SU($N$) symmetric fermions. All correlators are evaluated using the exact one-particle density matrix for $L = 15$ by setting $\mathbf{r}' = (0, R)$ and radius $R = 1$ at time $t = 0.022$. The color bar is non-linear by setting $\text{sgn}(G_{\text{R,C}})|G_{\text{R,C}}|^{1/4}$.

## 5 Conclusions and outlook

In this paper, we develop a theoretical framework to calculate the exact one-particle density matrix of $N$-component fermions in the limit of strong repulsion using a Bethe ansatz analysis working in the integrable regime of the SU($N$) Hubbard model. By splitting the problem into the spinless fermionic and SU($N$) Heisenberg models, we manage to compute these observables for a number of particles $N_p$, system size $L$ and number of components $N$ well beyond the current state-of-the-art tractable by numerical methods: on one hand, the numbers of particles and system size are well beyond exact diagonalization schemes; on the other hand, we remark that by Bethe ansatz we could access the limit of infinite repulsion that is a notoriously challenging limit for DMRG. On the technical side, we note that our Bethe ansatz scheme agrees well with the numerics (at least in the numbers which can be worked out) of the lattice model, also slightly beyond the dilute regime of Eq. (1). Specifically, we are able to calculate the correlations of systems composed of 38 sites and 12 particles for $N = 2$ and $N = 3$, with a total configuration space of 2 billion in the spinless configuration. Depending on $N$, this would correspond to a larger Hilbert space in the Hubbard model, such as $7.62 \times 10^{12}$ for $N = 2$. Exact diagonalization/Lanczos can only handle around 7 million. Therefore, there is no direct comparison between the two methods possible in this respect.

The Fourier transform of the correlation matrix is the momentum distribution $n(k)$ of the system. Despite being one of the simplest interesting correlation function, $n(k)$ reflects the many-body character of the quantum state. In particular, we quantify exactly the dependence of the gap at the Fermi point on different particle numbers and number of fermion components. We confirm the general expectation that for large number of components the Pauli exclusion principle relaxes. However, we find that the suppression of the gap for finite systems is non-monotonous.

We apply this scheme to the case in which SU($N$) matter can flow in ring-shaped potentials pierced by an effective magnetic flux $\phi$. As such, an additional complication in the calcula-

tion arises since the matter-wave states obey a complex dependence on $\phi$, ultimately leading to persistent currents with fractional quantization [41, 42]. In particular, we read-out such phenomenon in terms of *spin*-states of the Heisenberg SU($N$) model.

In this context, we give an example where the developed theory allows us to calculate readily available experimental observables such as time-of-flight measurements, both homodyne and self-heterodyne [39, 40].

We believe that our exact results can be exploited to benchmark the observables related to the one-body density matrix of SU($N$) fermions in the strongly interacting limiting. Finally, the theoretical framework we developed opens the possibility to study more complicated correlation functions.

# A  Appendix

In the following sections, we provide supporting details of the theory discussed in the manuscript.

## A.1  Separation of the spin and charge degrees of freedom

The one-dimensional SU(2) Hubbard Hamiltonian describing $N_p$ particles with $M$ flipped spins residing on a ring-shaped lattice with $L$ sites,

$$H = -t \sum_{j,\alpha} \left( c^\dagger_{j,\alpha} c_{j+1,\alpha} + \text{h.c.} \right) + U \sum_j n_{j,\uparrow} n_{j\downarrow}, \tag{A.1}$$

which is Bethe ansatz integrable. It was found that the eigenfunctions of the Hubbard model within a given sector $x_{Q1} \leq \ldots \leq x_{QN_p}$ are of the form

$$f(x_1,\ldots,x_{N_p}; \alpha_1,\ldots,\alpha_{N_p}) = \sum_P \text{sign}(PQ) \varphi_P(\alpha_{Q1},\ldots,\alpha_{QN_p}) \exp\left( i \sum_{j=1}^{N_p} k_{Pj} x_{Qj} \right), \tag{A.2}$$

where $P$ and $Q$ are permutations introduced to account for the eigenstates' dependence on the respective ordering of the electron coordinates $x_j$ and quasimomenta $k_j$, with $\varphi$ being the spin-dependent amplitude. The spin wavefunction contains all the spin configurations of the down spins can be expressed as

$$\varphi_P(\alpha_{Q1},\ldots,\alpha_{QN_p}) = \sum_v A_{\lambda_{v1},\ldots \lambda_{vM}} \prod_{l=1}^M F_P(\lambda_{vl}, y_l), \tag{A.3}$$

whereby we define

$$F_P(\lambda_{vl}, y) = \prod_{j=1}^{y-1} \frac{\sin k_{Pj} - \lambda_{vl} + i\frac{U}{4t}}{\sin k_{P(j+1)} - \lambda_{vl} - i\frac{U}{4t}}, \qquad A(\lambda_{v1},\ldots,\lambda_{vM}) = (-1)^v \prod_{i<j}^M \left( \Lambda_{vi} - \Lambda_{vj} - \iota\frac{U}{2} \right), \tag{A.4}$$

with $y$ corresponding to the coordinate of the electrons with spin-down in a given sector $Q$. As $U \to +\infty$, we can neglect the $\frac{\sin k_j}{U}$ terms such that

$$F(\lambda_{vl}, y) = \prod_{j=1}^{y-1} \frac{\lambda_{vl} - i\frac{U}{4t}}{\lambda_{vl} + i\frac{U}{4t}}. \tag{A.5}$$

After this treatment, the spin wavefunction is no longer dependent on the charge degrees of freedom through $F(\lambda, y)$. Consequently, the Bethe ansatz wavefunction as $U \to +\infty$ can be recast into the following form

$$f(x_1, \ldots, x_{N_p}; \alpha_1, \ldots, \alpha_{N_p}) = \operatorname{sign}(P)\operatorname{sign}(Q)\varphi(y_1, \ldots, y_M)\exp(ik_j x_{Q_j}). \qquad (A.6)$$

Additionally, we can go a step forward and show that in this limit the spin wavefunction corresponds to that of the one-dimensional anti-ferromagnetic SU(2) XXX Heisenberg chain. Indeed, it can be shown that

$$F(\lambda_{vl}, y) = \exp\left[iq_{vl}(y-1)\right], \qquad A(\lambda_{v1}, \ldots, \lambda_{vM}) = \exp\left[\frac{\iota}{2}\sum_{j<l}(\Psi_{vj,vl} - \pi)\right], \qquad (A.7)$$

by defining $q_\alpha = \pi + 2\arctan\left(4\frac{\Lambda_\alpha}{U}\right)$ and $\Psi_{\alpha,\beta} = \pi + 2\arctan\left(2\frac{\Lambda_\alpha - \Lambda_\beta}{U}\right)$. Consequently, the spin wavefunction becomes

$$\varphi(y_1, \ldots, y_M) = \sum_v \exp\left(\iota\sum_{l=1}^M q_{vl}y_l + \frac{\iota}{2}\sum_{j<k}\Psi_{vj,vk}\right), \qquad (A.8)$$

which except for a phase factor corresponds to the Bethe ansatz wavefunction of the Heisenberg model. Therefore, we have that

$$|\psi[U]\rangle_{\text{Fermi-Hubbard}} \xrightarrow{U \to +\infty} |\psi[\psi_{XXX}]\rangle_{\text{spinless}}. \qquad (A.9)$$

The same treatment can be applied for the SU($N$) Hubbard model, which results to be integrable in two limits: (i) large repulsive interactions $U >> t$ and filling fractions of one particle per site [48]; (ii) in the continuum limit of vanishing lattice spacing achievable by dilute filling fractions [22,31]. The Bethe ansatz wavefunction for the model is of the same form as the one outlined in Equation (A.2) with the added difference that the $\varphi$ houses the extra spin degrees of freedom. In the following, we will focus on the second integrable regime and illustrate the decoupling of the spin and charge degrees of freedom for SU($N$) fermions through the Bethe ansatz equations.

### A.1.1   Extension to SU($N$) fermions

In the continuous limit, the SU($N$) Hubbard model tends to the Gaudin-Yang-Sutherland Hamiltonian describing $N$-component fermions with a delta potential interaction [31, 38], which reads

$$\mathcal{H}_{\text{GYS}} = \sum_{m=1}^N \sum_{i=1}^{N_m}\left(-i\frac{\partial}{\partial x_{i,m}} - \frac{2\pi}{L_R}\phi\right)^2 + 4U\sum_{m<n}^N \sum_{i,j}\delta(x_{i,m} - x_{j,n}), \qquad (A.10)$$

where $N_m$ is the number of electrons with colour $\alpha$ of with $m = 1, \ldots N$, $L_R$ being the size of the ring and $\phi$ denoting the effective magnetic flux threading the system.
The Bethe ansatz equations for the model are as follows,

$$e^{i(k_j L - \phi)} = \prod_{\alpha=1}^{M_1}\frac{4\left(k_j - \lambda_\alpha^{(1)}\right) + iU}{4\left(k_j - \lambda_\alpha^{(1)}\right) - iU}, \quad j = 1, \ldots, N_p, \qquad (A.11)$$

$$\prod_{\beta\neq\alpha}^{M_r}\frac{2\big(\lambda_\alpha^{(r)}-\lambda_\beta^{(r)}\big)+iU}{2\big(\lambda_\alpha^{(r)}-\lambda_\beta^{(r)}\big)-iU}=\prod_{\beta=1}^{M_{r-1}}\frac{4\big(\lambda_\alpha^{(r)}-\lambda_\beta^{(r-1)}\big)+iU}{4\big(\lambda_\alpha^{(r)}-\lambda_\beta^{(r-1)}\big)-iU}\cdot\prod_{\beta=1}^{M_{r+1}}\frac{4\big(\lambda_\alpha^{(r)}-\lambda_\beta^{(r+1)}\big)+iU}{4\big(\lambda_\alpha^{(r)}-\lambda_\beta^{(r+1)}\big)-iU}\,,\quad \alpha=1,\ldots,M_r,$$

(A.12)

for $r=1,\ldots,N-1$ where $M_0=N_p$, $M_N=0$ and $\lambda_\beta^{(0)}=k_\beta$. $N_p$ denotes the number of particles, $M_r$ corresponds to the colour with $k_j$ and $\lambda_\alpha^{(r)}$ being the charge and spin momenta respectively. The energy corresponding to the state for every solution of these equations is $E=\sum_j^{N_p}k_j^2$.

For SU(3) fermions, one obtains the three nested non-linear equations

$$e^{i(k_jL-\phi)}=\prod_{\alpha=1}^{M_1}\frac{4(k_j-q_\alpha)+iU}{4(k_j-q_\alpha)-iU}\,,$$

(A.13)

$$\prod_{\beta\neq\alpha}^{M_1}\frac{2(q_\alpha-q_\beta)+iU}{2(q_\alpha-q_\beta)-iU}=\prod_{j=1}^{N_p}\frac{4(q_\alpha-k_j)+iU}{4(q_\alpha-k_j)-iU}\prod_{a=1}^{M_2}\frac{4(q_\alpha-p_a)+iU}{4(q_\alpha-p_a)-iU}\,,$$

(A.14)

$$\prod_{b\neq a}^{M_2}\frac{2(p_\alpha-p_\beta)+iU}{2(p_\alpha-p_\beta)-iU}=\prod_{\beta=1}^{M_1}\frac{4(p_\alpha-q_\beta)+iU}{4(p_\alpha-q_\beta)-iU}\,,$$

(A.15)

where $\lambda_\beta^{(1)}$ and $\lambda_\beta^{(2)}$ were changed to $q_\beta$ and $p_a$ for the sake of convenience. In the limit $U\to\infty$ [33, 41, 42] we observe that $k_j/U$ will tend to zero, since all $k$ of the ground-state are real here for repulsive $U$. Consequently, the Bethe equations read

$$e^{i(k_jL-\phi)}=\prod_{\alpha=1}^{M_1}\frac{2Q_\alpha-i}{2Q_\alpha+i}\,,$$

(A.16)

$$\prod_{\beta\neq\alpha}^{M_1}\frac{(Q_\alpha-Q_\beta)+i}{(Q_\alpha-Q_\beta)-i}=\left[\frac{2Q_\alpha+i}{2Q_\alpha-i}\right]^{N_p}\prod_{a=1}^{M_2}\frac{2(Q_\alpha-P_a)+i}{2(Q_\alpha-P_a)-i}\,,$$

(A.17)

$$\prod_{b\neq a}^{M_2}\frac{P_a-P_a+i}{P_a-P_b-i}=\prod_{\beta=1}^{M_1}\frac{2(P_a-Q_\beta)+i}{2(P_a-Q_\beta)-i}\,,$$

(A.18)

defining $Q_\alpha=2\frac{q_\alpha}{U}$ and $P_a=2\frac{p_a}{U}$ respectively. The Bethe equations decouple into that of a model of spinless fermions (A.16) and those of an SU(3) Heisenberg magnet (A.17) and (A.18).

Subsequently, by taking the logarithm of Equations (A.16) through (A.18) and using

$$2i\arctan x=\pm\pi+\ln\frac{x-i}{x+i}\,,$$

(A.19)

it can be shown that the quasimomenta $k_j$ can be expressed as [42]

$$k_j=\frac{2\pi}{L}\left[I_j+\frac{1}{N_p}\left(\sum_{\alpha=1}^{M_1}J_\alpha+\sum_{a=1}^{M_2}L_a\right)+\phi\right]\,,$$

(A.20)

in terms of the charge $I_j$ and two sets of spin $J_\alpha$, $L_a$ quantum numbers. By exploiting different configurations of these quantum numbers, we can construct all the excitations and the corresponding Bethe ansatz wavefunction. The procedure outlined here holds for any $N$-component

fermionic systems, with the added difference that there will be $N-1$ sets of spin quantum numbers (2 for the considered SU(3) case).

For strong repulsive couplings, the ground-state energy of the Hubbard model fractionalizes with a reduced period of $\frac{1}{N_p}$ as a combined effect of the effective magnetic flux, interaction strength and spin correlations [41,42], which is in turn reflected in the momentum distribution [37]. In the Bethe ansatz language, this phenomenon is accounted for through various configurations of the spin quantum numbers $X$ that correspond to different spin excitations that are generated in the ground-state to counteract the increase in the flux.

### A.2  SU($N$) Heisenberg model

The SU(2) Heisenberg model is a sum of permutation operators

$$\mathcal{H}_{XXX} = \sum_i^{N_p} P_{i,i+1} = \sum_i^{N_p} (\mathbb{1} + \vec{\sigma}_{i+1} \cdot \vec{\sigma}_i)/2, \tag{A.21}$$

with $\vec{\sigma}_i$ corresponding to the Pauli matrices, the three generators of the SU(2) Lie algebra. In the case of the SU($N$) Heisenberg model, the Hamiltonian can be constructed in a similar fashion [22,48]. In general we obtain for the generators $\lambda_i$ of the SU($N$)

$$P_{i,i+1} = \frac{1}{N}\mathbb{1} + \frac{1}{2}\vec{\lambda}_i \cdot \vec{\lambda}_{i+1}, \tag{A.22}$$

which acts on sites $i$ and $i+1$ permuting the SU($N$) states.

#### A.2.1  Details about the SU($N$) generators

The generators in the Lie algebra of SU($N$) are analogues of the Pauli matrices in SU(2). Taking SU(3) as an example, we have six non-diagonal generators

$$\lambda_1 = \begin{pmatrix} 0 & 1 & 0 \\ 1 & 0 & 0 \\ 0 & 0 & 0 \end{pmatrix}, \quad \lambda_2 = \begin{pmatrix} 0 & -i & 0 \\ i & 0 & 0 \\ 0 & 0 & 0 \end{pmatrix}, \quad \lambda_3 = \begin{pmatrix} 0 & 0 & 1 \\ 0 & 0 & 0 \\ 1 & 0 & 0 \end{pmatrix},$$

$$\lambda_4 = \begin{pmatrix} 0 & 0 & -i \\ 0 & 0 & 0 \\ i & 0 & 0 \end{pmatrix}, \quad \lambda_5 = \begin{pmatrix} 0 & 0 & 0 \\ 0 & 0 & 1 \\ 0 & 1 & 0 \end{pmatrix}, \quad \lambda_6 = \begin{pmatrix} 0 & 0 & 0 \\ 0 & 0 & -i \\ 0 & i & 0 \end{pmatrix}, \tag{A.23}$$

that together with two diagonal generators

$$\lambda_7 = \begin{pmatrix} 1 & 0 & 0 \\ 0 & -1 & 0 \\ 0 & 0 & 0 \end{pmatrix}, \quad \lambda_8 = \frac{1}{\sqrt{3}}\begin{pmatrix} 1 & 0 & 0 \\ 0 & 1 & 0 \\ 0 & 0 & -2 \end{pmatrix}, \tag{A.24}$$

comprise the Gell-Mann matrices that are the matrix representation of the SU(3) Lie algebra. For generalization purposes, the generators were grouped by defining $\lambda_{2p-1/2p}$, $p = 1, \ldots, \frac{N(N-1)}{2}$ which are analogues to the $\sigma_{x/y}$ that operate between the different subspaces of SU(3) which are $(i,j)$, $i < j$. Here, both run from 1 to 3. We decided to group the elements of the diagonal Cartan basis at the end as $\lambda_7$ and $\lambda_8$, which differs from the standard Gell-Mann matrices, but is eases the generalisation. For the extension to $SU(N)$, one has to consider the $N(N-1)/2$ elements $\lambda_i$, which would correspond to $\sigma_{x/y}$ in some space $(i,j)$, where $i < j \in \{1, \ldots, N\}$. Additionally, the corresponding diagonal Cartan elements need to be taken into account. There are $N-1$ Cartan elements that can be constructed via the following formula $\lambda_{N^2-(N+1)+m} = \text{diag}\,\{1, \ldots, 1, -(m-1), 0, \ldots, 0\}/\sqrt{m(m-1)/2}$ where $m = 2, \cdots, N$; the $1/0$ occurs $(m-1)/(N-m)$ times, respectively.

### A.2.2 Casimirs of SU($N$) fermions

Whereas in SU(2) we have a single Casimir operator, for SU($N$) we are faced with $N-1$ Casimirs. Out of these Casimirs, we are only interested in the quadratic Casimir, which for the fundamental representation reads

$$C_1 = \frac{1}{4} \sum_{i=1}^{N^2-1} \lambda_i^2, \tag{A.25}$$

as it relates to the total spin quantum number $\vec{S}^2$, which is necessary for us to classify the Heisenberg eigenstates. To this end we have to evaluate the Casimir in various SU($N$) representations. In the following, we sketch the procedure to write the quadratic Casimir operator for SU(3) and SU(4).

We start by looking at the SU(3) case, where its representations $\Lambda(n_1, n_2)$ are labeled by integer numbers which correspond to the simple Cartan elements $(h_1, h_2)$: $\Lambda(n_1, n_2) = \vec{n} \cdot \vec{h}$. The elements are given by

$$h_1 = (\lambda_3, \lambda_8) \cdot (1, 0)^T \implies h_1 = (\sigma_z)_{1,2}; \quad \vec{h}_1 := (1, 0), \tag{A.26}$$

$$h_2 = (\lambda_3, \lambda_8) \cdot \left(-\frac{1}{2}, \frac{\sqrt{3}}{2}\right)^T \implies h_2 = (\sigma_z)_{2,3}; \quad \vec{h}_2 := (-1, \sqrt{3})/2. \tag{A.27}$$

To calculate the quadratic Casimir values for these representations $\Lambda$, we need the Cartan matrix

$$C_h = 2\left(\frac{\vec{h}_i \cdot \vec{h}_j}{\|h_i\|^2}\right)_{ij} = \begin{pmatrix} 2 & -1 \\ -1 & 2 \end{pmatrix}, \tag{A.28}$$

defined using the Killing form $(\lambda_j, \lambda_k) := K(\lambda_j, \lambda_k) = \frac{1}{8}\text{tr}\,\lambda_j\lambda_k = \frac{1}{4}\delta_{jk}$ (see [57], chapter 12 for the evaluation of the Casimir). We obtain

$$3C_1 = (\Lambda, \Lambda + \delta) = \left[\vec{n}C_h^{-1} + \vec{\delta}\right]\vec{n}^T = \sum_{i=1}^{2} n_i(n_i + 3) + n_1 n_2, \tag{A.29}$$

giving the value of 4/3 for the fundamental representations $(1, 0)$ and $(0, 1)$. Here, $\vec{\delta} = \frac{1}{2}\sum_{h \in \Delta_+} \vec{h} = (2, 2)$ (see [57–59]) for the positive roots $\Delta_+$. These are the two simple roots together with their sum, $h_1 + h_2$. If one introduces half-integer values as in the SU(2) representation for each $n_i$ such that $(n_i = 2J_i)$, we obtain

$$C_1(\Lambda) = \frac{4}{3}\left[\sum_i J_i\left(J_i + \frac{3}{2}\right) + J_1 J_2\right]. \tag{A.30}$$

Likewise for SU(4), the representations $\Lambda(n_1, n_2, n_3)$ of SU(4) are labeled by the Cartan elements $(h_1, h_2, h_3)$: $\Lambda(n_1, n_2, n_3) = \vec{n} \cdot \vec{h}$, which are given by

$$h_1 = (\lambda_{13}, \lambda_{14}, \lambda_{15}) \cdot (1, 0, 0)^T \implies h_1 = (\sigma_z)_{1,2}; \quad \vec{h}_1 := (1, 0, 0), \tag{A.31}$$

$$h_2 = (\lambda_{13}, \lambda_{14}, \lambda_{15}) \cdot \left(-\frac{1}{2}, \frac{\sqrt{3}}{2}, 0\right)^T \implies h_2 = (\sigma_z)_{2,3}; \quad \vec{h}_2 := (-1, \sqrt{3}, 0)/2, \tag{A.32}$$

$$h_3 = (\lambda_{13}, \lambda_{14}, \lambda_{15}) \cdot \left(0, -\frac{1}{\sqrt{3}}, \sqrt{\frac{2}{3}}\right)^T \implies h_3 = (\sigma_z)_{3,4}; \quad \vec{h}_3 := (0, -1, \sqrt{2})/\sqrt{3}. \tag{A.33}$$

The corresponding Cartan matrix reads

$$C_h = \begin{pmatrix} 2 & -1 & 0 \\ -1 & 2 & -1 \\ 0 & -1 & 2 \end{pmatrix}. \tag{A.34}$$

Upon evaluating the quadratic Casimir as in Equation (A.30), we have that

$$2C_1(\Lambda) = (n_1 + 2n_2 + n_3)^2 + n_1\left(2n_1 + \frac{3}{4}\right) + n_3\left(2n_3 + \frac{3}{4}\right) + n_2, \tag{A.35}$$

with $\vec{\delta} = \frac{1}{2}\sum_{h\in\Delta_+}\vec{h} = (3,4,3)$. Here, the positive roots are the three simple roots together with $h_1 + h_2$, $h_2 + h_3$, and $h_1 + h_2 + h_3$. Introducing half-integer values as for SU(2), we obtain

$$C_1(\Lambda) = 2J_2\left(J_2 + \frac{1}{4}\right) + \sum_{i=1,3} 3J_i\left(2J_i + \frac{1}{4}\right) + 4\sum_{i<j} J_i J_j, \tag{A.36}$$

leading to the value of 15/8 for the fundamental representations $(1,0,0)$ and $(0,0,1)$, and 5/4 for the representation $(0,1,0)$.

## A.3 Evaluating correlation functions

In the previous sections, we outlined how the spin and charge degrees of freedom decouple yielding a simplified form of the Bethe ansatz wavefunction (A.37), that at infinite repulsion reads

$$f(x_1, \ldots, x_{N_p}; \alpha_1, \ldots, \alpha_{N_p}) = \text{sign}(Q)\det[\exp(ik_j x_l)]_{jl} \varphi(y_1, \ldots, y_M). \tag{A.37}$$

Here, we are going to show how to evaluate the Slater determinant of the charge degrees of freedom and the corresponding spin wavefunction in the presence of an effective magnetic flux.

### A.3.1 Slater determinant

To calculate the Slater determinant of spinless fermions, we need to start by noting that

$$k_j = -(N_p - 1 + \ell)\frac{\pi}{L} + (j-1)\Delta k + k_0 + \frac{X}{N_p}, \quad j = 1, \ldots, N_p, \tag{A.38}$$

where $\Delta k = \frac{2\pi}{L}$, $X$ denotes the sum over the spin quantum numbers and $\ell$ is the angular momentum. $k_0$ is a constant shift can be 0 or $-\frac{\pi}{L}$ for systems with $(2m)N$ and $(2m+1)N$ fermions respectively, that will henceforth be termed as paramagnetic and diamagnetic. Through Equation (A.38), we can re-write the Slater determinant in the following form

$$\det[\exp(ik_j x_l)]_{jl} = \exp(ik_1 r_{cm} N_p)\det\begin{pmatrix} 1 & y_1 & y_1^2 & \cdots & y_1^{N_p-1} \\ 1 & y_2 & y_2^2 & \cdots & y_2^{N_p-1} \\ 1 & y_3 & y_3^2 & \cdots & y_3^{N_p-1} \\ \vdots & \vdots & \vdots & \ddots & \vdots \\ 1 & y_{N_p} & y_{N_p}^2 & \cdots & y_{N_p}^{N_p-1} \end{pmatrix}, \tag{A.39}$$

with $r_{cm}$ denoting $\sum_i x_i/N_p$ which we refer to as the *center of mass*. The matrix elements of the determinant are of the form $y_m^{j-l} = \exp(i(k_j - k_l)r_m)$, whereby we made use of the fact

that all the quasimomenta are equidistant. By noting that the matrix in Equation (A.39) has the same structure of the Vandermonde matrix [33], we can express the Slater determinant as

$$\det[\exp(ik_j x_{Qj})] = \exp(ik_1 r_{cm} N_p) \prod_{1 \le i < j \le n} (\exp(i\Delta k r_j) - \exp(i\Delta k r_i)), \quad (A.40)$$

which upon simplification reads

$$\det\left[\exp(ik_j x_{Qj})\right] = \exp(ik_1 r_{cm} N_p) \prod_{1 \le i < j \le n} \exp\left(i\Delta k \frac{r_j + r_i}{2}\right) \prod_{1 \le i < j \le n} \left(2i \sin \frac{\Delta k(r_j - r_i)}{2}\right). \quad (A.41)$$

This expression can be further simplified by noticing that

$$\prod_{1 \le i < j \le n} \exp\left(i\Delta k \frac{r_j + r_i}{2}\right) = \exp\left(\frac{i\Delta k}{2}\left[r_{cm} N_p^2 - r_{cm} N_p\right]\right), \quad (A.42)$$

that in conjunction with Equation (A.38) reduces Equation (A.40) into

$$\det\left[\exp(ik_j x_{Qj})\right] = \exp\left(i\left[k_0 + \frac{X}{N_p} - \ell \Delta k\right] r_{cm} N_p\right) \prod_{1 \le i < j \le n} \left(2i \sin \frac{\Delta k(r_j - r_i)}{2}\right). \quad (A.43)$$

In the presence of an effective magnetic flux, the variables $X$ and $\ell$ need to be changed in order to counteract the increase in flux. For the spin quantum numbers, the shift needs to satisfy the degeneracy point equation [41,42]

$$\frac{2w-1}{2N_p} \le \phi + D \le \frac{2w+1}{2N_p}, \quad \text{where} \quad X = -w, \quad (A.44)$$

with $\phi$ ranging from 0.0 to 1.0 and $D$ being $0 \left[-\frac{1}{2}\right]$ for diamagnetic [paramagnetic] systems. Upon increasing $\phi$, the angular momentum of the system increases at $\phi = \left(s \pm \frac{1}{2N_p} + \delta\right)$ with $s$ being (half-odd) integer in the case of (diamagnetic) paramagnetic systems, with $\delta = \mp \frac{1}{2N_p}$ for an odd number of particles.

### A.3.2 Resolving degeneracies of the spin wavefunction

As $U \to +\infty$, all the spin configurations of the model are degenerate. The reason is that the energy contribution from the spin part of the wavefunction $E_{\text{spin}}$ is of the order $\frac{t}{U}$. However, there is no spin degeneracy observed in the Hubbard model; the ground-state is non-degenerate for SU(2), except for special points in flux with an eigenenergy crossing. Hence, a single state has to be chosen properly for matching with the Hubbard model. Due to the symmetry of both models, we choose the square of the total spin, $\vec{S}_{\text{tot}}^2$, or quadratic Casimir operator $C_1$ to label the eigenstates. The selected eigenstates of both models need to have the same value for this operator. We used this benchmarking with the Hubbard model only for small system sizes in order to understand what are the representations of the Heisenberg model we have to choose.

We observe that the resulting composition from spinless Fermions and Heisenberg Hamiltonian results in a translationally invariant model only in cases where these states match. We use this as a control mechanism. As already explained in the main text, the spin wavefunction $\varphi(y_1, \ldots, y_M)$ is obtained by performing exact diagonalization resp. Lanczos methods of the one-dimensional anti-ferromagnetic Heisenberg model.

*a) Zero flux* – The ground-state with odd and even number of particles per species for the Hubbard model corresponds to different values of the Casimir operator, and therefore to different

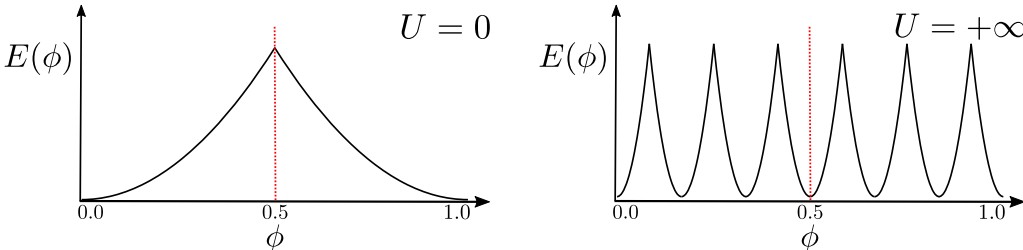

Figure 7: Schematic figure of the energy $E(\phi)$ against effective magnetic flux $\phi$ for $N_p = 6$ particles. Left panel displays the energy landscape at $U = 0$ while right panel shows the $U \to +\infty$ case where the parabolas are fractionalized [37].

representations of the SU($N$) algebra. For odd number of particles per species $N_p/N$, it corresponds to a singlet state for all SU($N$). The ground-state of the anti-ferromagnetic Heisenberg model instead is always a singlet and non-degenerate for all SU($N$). Therefore, we choose this state as the lowest energy eigenstate of the Heisenberg model with this property for $U \to \infty$ for an odd occupation number per species.

For an even number of particles per species, we have to choose a different state. For the SU(2) this is the lowest non-degenerate excited triplet-state (of total spin $J = 1$, $\vec{J}^2 = J(J+1)$) in the spectrum of the Heisenberg model. It corresponds to an $n = 2J$-representation (see section A.2.2 of the Appendix). For SU($N > 2$), i.e. $N = 3$ and $N = 4$, it is the first non-degenerate state with Casimir eigenvalue $C_1 = 6$. Examples are the 10-dimensional representations $(n_1, n_2) = (3, 0)$ for SU(3) and correspondingly $(n_1, n_2, n_3) = (4, 0, 0)$ for SU(4). The numbers $n_i$ in the SU(3) representations correspond to the numbers $p$ and $q$ frequently used in SU(3) representations in the mathematical literature or high energy physics; there they represent the number of (anti-)quarks. The dimension of a representation $(n_1, n_2)$ of SU(3) is $d(n_1, n_2) = (n_1 + 1)(n_2 + 1)(n_1 + n_2 + 2)/2$. Both representations for SU(3) and SU(4) have a Casimir value of $C_1 = 6$. We assume that this representation will be $(N, 0, \ldots, 0)$ for SU($N$). This state takes the role of the non-degenerate triplet state of SU(2) in the zero field ground-state for an even species number occupation.

*b) Non-zero flux–* The analysis of non-zero magnetic flux is motivated from an atomtronics context [28, 37]. As mentioned previously, for strong repulsive interactions a fractionalization of the persistent currents in the model is observed [41, 42]. Figure 7 shows an example of the change of the energy landscape when going from non-interacting to strongly interacting particles in presence of an effective magnetic flux. This fractionalization appears since formerly higher excited states are bent by the field to be the ground-state. A unique method to identify these states would be to utilize the SU($N$) Heisenberg Bethe equations, which need to have the same spin quantum number configurations as their Hubbard counterparts. In this manner, we are guaranteed that the corresponding eigenstates obtained from the Heisenberg model correspond to the ground-state of the Hubbard model. However, it is rather tedious to achieve the whole state using this method. This is particularly true because the Bethe ansatz gives direct solutions only for the highest weight states and we work at an equal occupation of each species: the resulting state is then obtained by applying sufficiently often the proper lowering operators of SU($N$).

In the case of paramagnetic systems, i.e. an even number of particles per species, the central fractionalized parabola (centered around $\phi = 0.5$), corresponds to a singlet state. This parabola results to be non-degenerate for the Heisenberg model and is therefore easily distinguished. As such, one obtains the corresponding states for the outer and central fractionalized

parabolas in a straight forward manner for arbitrary SU($N$). We mention though that in order to find the corresponding state for the outer parabola and the paramagnetic case (even occupation of each species), we have to single out a non-degenerate excited state with Casimir value $C_1 = 6$.

For finite field and degenerate ground-states of the Hubbard model, we do not have a general procedure to choose the states for SU($N > 2$). Therefore, we explain our approach in considering SU(2) first and then apply it to SU(3) symmetric fermions.

In the case of SU(2), the remaining fractionalized parabolas (i.e. excluding the two outer parabolas and the central one) have a common spin value of $J = 1$. This in turn results in a two-fold degeneracy in the spin-$\frac{1}{2}$ Heisenberg model for a given collective spin quantum number $|X|$. Hence, the relevant states for two of the parabolas of a given $|X|$ are superpositions of these degenerate eigenstates of the Heisenberg Hamiltonian. These states can be separated by different eigenvalues for $P_{j,j-1}$, part of the Heisenberg model but not commuting with it. We call both eigenstates of this permutation operator $|\psi_{1/2}\rangle$. The states $|\psi_{l/r}\rangle$ corresponding to the inner branches of the fractionalization are obtained from the two spin- and energy-degenerate states $|\psi_{1/2}\rangle$ as

$$|\psi_{l/r}\rangle := \frac{1}{\sqrt{2}} \left( |\psi_1\rangle \pm i |\psi_2\rangle \right). \tag{A.45}$$

It is worth mentioning that these states correspond to fractionalized parabolas that emerge from a singlet state in the absence of flux to a non-degenerate triplet state with each of the basis elements being non-zero. This happens here gradually via intermediate triplet states where certain basis states are excluded. As an example, we take an SU(2) state with 6 particles to explain this better. Since this state has 3 particles of each species ($\uparrow$ or $\downarrow$) the parabolas start from a singlet and persist as a triplet state during their fractionalization up to the center parabola.

The singlet state is made of three distinct configurations: a) $|111000\rangle \pm$ cyclic permutations, b) $|101010\rangle - |010101\rangle$, and c) the possible remaining configurations with alternating sign (singlet state). This is mediated via fractionalized states where the component a) is missing in the first inner parabola and additionally the component b) vanishes for the second parabola. The triplet of the central parabola has the same components as the singlet state but without alternating signs.

For SU(2) the corresponding states that belong to the fractionalized parabolas have been triplet states, as was the non-degenerate state corresponding to either the central (diamagnetic, odd species number) or the outer parabola (paramagnetic, even species number). However, the representations are modified for $N > 2$ in the intermediate parabolas. In the case of SU(3) we obtain $(n_1, n_2) = (1, 1)$ as the 8-dimensional representation (instead of $(n_1, n_2) = (3, 0)$) which governs the intermediate parabola. For SU(4) it is $(1, 2, 0)$ instead of $(4, 0, 0)$ (see Appendix A.2.2). The Casimir $C_1$ has values 3 and 4 respectively. These representations take the role of the degenerate triplet state of SU(2).

In the case non-vanishing flux threading a ring of SU($N > 2$) symmetric fermions, the ground-states of the Hubbard Hamiltonian (1) belonging to a given $|X|$, is $N-1$-fold degenerate coming from the $N-1$ sets of spin quantum numbers. This degeneracy holds for the inner fractionalized parabolas. As a consequence of its one-to-one correspondence with the Hubbard model, these degeneracies are manifested in the Heisenberg model, in addition to the two-fold degeneracy mentioned previously for both parabolas with equal values for $|X|$. In order for this extra

degeneracy to be resolved, we make certain coefficients of the wavefunction in the Heisenberg basis vanish by according superpositions of the degenerate states. This has been motivated by former observations in SU(2) (see discussion above).

To get a better idea of how this is done explicitly, here we exemplify on the case of 3 particles in SU(3). There are only two possible values for $|X|$ in this case and each parabola is two-fold degenerate in the Hubbard model. The degeneracy of the Heisenberg model is hence 4-fold. So, the distinct states have to be selected from a remaining two-fold degeneracy of the operator $P_{i,i+1}$. The zeroth parabola is in the singlet state of SU(3) that belongs to $C_1 = 0$ for which every component of the wavefunction is non-zero. Both two-fold degenerate inner parabolas have $C_1 = 3$ and correspond in one case to the positive or negative permutation of the species number only; in the second degenerate case they correspond to configurations $\{|021\rangle, |102\rangle\}$ and $\{|120\rangle, |201\rangle\}$ as the only non-zero component. These are the states $\{|\psi_1\rangle, |\psi_2\rangle\}$ that are to be superposed by formula (A.45). The direct way to obtain the corresponding state of the Heisenberg model is via the Bethe ansatz wavefunction for the same spin quantum numbers of the Hubbard model. The degeneracies amount to $2(N-1)$-fold for the SU($N$) Heisenberg model. These are distinguished by the eigenstates of the permutation operator $P_{j,j+1}$ up to a remaining $(N-1)$-fold degeneracy.

## A.4  Comparison with numerics

In this section, we compare the correlations obtained via the method presented in this paper to those obtained through exact diagonalization using the Lanczos algorithm. The error between the two methods is estimated by calculating the relative correlation distance $\mathcal{D}$ for the ground state, which is defined as

$$\mathcal{D} = \frac{1}{L}\sqrt{\sum_i \left(c_{1,i}^{(ED)} - c_{1,i}\right)^2},\qquad(A.46)$$

where $c^{(ED)}$ correspond to the correlations obtained through exact diagonalization. We note that because of the periodicity of the system all $c_{i,j}$ are a circular shift of $c_{1,j}$, such that we only need to sum once. Some of the comparisons that were carried out are tabulated in Table 1. Naturally, we find that as one goes to large interactions, the agreement of the correlations between the two methods increases. Such a result is to be expected as our proposed scheme is viable in the limit of infinite repulsion. Furthermore, we highlight that our system is far from being in the dilute limit, which is one of the integrable regimes of the Hubbard model. In spite of this, there is an excellent agreement between exact diagonalization and our scheme that is intrisically reliant on the system being Bethe ansatz integrable. Bethe ansatz integrability hinges on the fact that the scattering of more than two particles does not occur (Yang-Baxter factorization of the scattering matrix). In the infinite repulsive regime, the multiparticle scattering is suppressed since the probability of two particles interacting is vanishing. Therefore, despite the fact that we are far from the dilute limit condition, the system is indeed very close to be integrable for low lying states, and our method is able to accurately tackle the infinite repulsive limit of the SU($N$) Hubbard model.

The Hilbert space of the Hubbard model for an equal number of particles per colour is given by $\binom{L}{N_c}^N$, where $L$ corresponds to the system size, $N_c$ to the number of particles in a given colour, and $N$ is the number of components. It is straightforward to see that the size of the Hilbert space increases at least exponentially on going to a larger value of any of these three variables. When it comes to exact diagonalization, the size of the Hilbert space is one of the limitations as it exceeds the memory of the computer defined as Msize. This can be estimated in the following manner $\text{Msize[GByte]} = \frac{\binom{L}{N_c}^N \times N_p \times 64}{1024 \times 1024 \times 1024 \times 8}$, where we count the number of

Table 1: The relative correlation distance $\mathcal{D}$, defined in Equation (A.46), is presented as a function of the number of sites $L$, particles $N_p$, components $N$, and interaction $U^{(\mathrm{ED})}$. This interaction corresponds to that used in exact diagonalization simulations, whilst that of our scheme is always infinite.

| $L$ | $N_p$ | $N$ | $U^{(\mathrm{ED})}$ | $\mathcal{D}$ |
|---|---|---|---|---|
| 15 | 4 | 2 | 750 | $2.68 \times 10^{-5}$ |
| 15 | 4 | 2 | 10,000 | $2.01 \times 10^{-6}$ |
| 15 | 6 | 2 | 750 | $6.65 \times 10^{-5}$ |
| 15 | 6 | 2 | 5000 | $9.99 \times 10^{-6}$ |
| 15 | 3 | 3 | 1000 | $1.60 \times 10^{-5}$ |
| 15 | 3 | 3 | 5000 | $3.19 \times 10^{-6}$ |
| 10 | 6 | 3 | 1000 | $8.34 \times 10^{-5}$ |
| 10 | 6 | 3 | 5000 | $1.67 \times 10^{-5}$ |

configurations, the number of particles (that gives the numbers we need to store) and the bits occupy by Int64, then we convert this into GigaBytes. Specifically, through our scheme we are able to consider systems with $N_p = 12$, $L = 38$ and $N = 3$, which correspond to a Msize= 242 GB for the spinless, and 3500 TB for the corresponding Hubbard model, which is clearly not attainable in current High Performance Computing systems. However, in our case we can perform calculations without storing the configurations. Similar approaches can also be followed in exact diagonalization, but not with these parameters. In the current state-of-the-art, one can diagonalize a Hilbert space of around 7 million (corresponding to Msize=0.31GB) using the Lanczos algorithm when taking into account the large matrices and values of the interactions used in the numerical operations, such as for example the calculation of correlations. Our proposed scheme is able to go to larger system parameters on account of the spin-charge decoupling.

By separating the problem into the spinless and Heisenberg parts, we deal with small Hilbert space dimensions that are given by $\binom{L}{N_p}$ and $\frac{N_p!}{(N_c!)^N}$ respectively. In doing so, the size of the system that we can consider, i.e. the number of sites, comes down to calculating the Slater determinant (A.43). Such a calculation is limited not by the memory size but by its runtime. However, in this manner one can calculate large system sizes, such as for example 38 sites that corresponds to a Hilbert space of around 2 billion for the spinless part. The other part of the problem lies in diagonalizing the Heisenberg matrix, whose dimensions are significantly smaller than its Hubbard counterpart, enabling us to calculate the system parameters displayed in this manuscript. It should be stressed that even through this scheme one is not able to calculate systems with very large particle numbers, as this part of the calculation is still affected by the dimensions of the matrix under consideration. Additionally, for $N_p = (2m)N$ we need to consider the excited states of the Heisenberg model in order to get the actual ground-state of the Hubbard system, which means that we need to perform the full diagonalization of the former instead of employing the lanczos algorithm.

Lastly, we close this section by drawing comparisons with DMRG. The system under consideration is infinitely repulsive SU($N$) fermions residing on a ring. In this context, DMRG has problems with convergence due to the large repulsive interactions and the high number of degeneracies present in the system. It is also limited by the periodic boundary conditions. Nonetheless, as we remarked in the manuscript, in the regime of intermediate interactions, DMRG can still be employed giving a good agreement with exact diagonalization and the proposed scheme.

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
