# Peer review of "Exact one-particle density matrix for SU(N ) fermionic matter-waves in the strong repulsive limit"

_SciPost Physics Core, doi:SciPost Phys. 15, 006 (2023)_

## Round 1 · Referee Report · Anonymous (Referee 1) · 2023-1-22

Strengths

The authors claimed a new method which was built to evaluate the one-particle correlation function of SU(N) fermionic systems. They carried out some calculation of the one-particle density matrix in mesoscopic systems with the particle number less than 10. This research topic is of great interest and importance.

Weaknesses

This paper did not reach a satisfactory level of research. Their main result was just a little extension of the method used in Ref.[33], and their previous study Ref.[42] etc. The paper lacks essential new results of multicommponent Fermi gases /Hubbard model.

Report

The authors claimed a new method which was built to evaluate the one-particle correlation function of SU(N) fermionic systems. They carried out some calculation of such one-particle density matrix in mesoscopic systems with the particle number less than 10. Although this research topic is of great interest and importance, to my best knowledge, this paper did not reach a satisfactory level of research. Their main result was just a little extension of the method used in Ref.[33], and the previous paper Ref.[42] etc. Here I raise up few questions for their manuscript:

(1) The authors claimed their results to be exact. It is true only if their calculations are efficient and correct in terms of Bethe ansatz equations within strong coupling limit. According to their statement, ‘dilute limit’ (Np/L <<1) was necessarily needed. However, in Figs.4 & 6, there are cases that 10 particle on 15 sites. This density is far from the dilute limit, and they did not provide verifications or comparisons with the results based on other methods such as exactly diagonalization (ED) or DMRG. Therefore, it should be of caution with the validity of their results in these Figures.

(2) They claimed the methods are applicable to SU(N) fermionic system. Obviously, in their main results i.e. Figs.4 & 6, only situations of SU(2) were considered. In fact, for SU(2) fermionic system in strong coupling limit, various methods and results have been obtained in literature, see [33], also see the review articles in this filed, etc. The discussion showed in Fig.5 looks rather rough and lacks rigorousness. The important references for the study of the SU(N) interacting fermions in 1D are missing.

(3) The authors claimed their results are well-beyond ED, while the manuscript lacks of explicit proof or comparison with ED results.

(4) Their results did not show essential physics of SU(N) fermions, such as universal nature of momentum distributions in connection to the Tan’s Contact, as well as to the power law decay of Luttinger liquid, etc. In general, when the component N>>1, the behaviour of ground properties of the SU(N) fermions coincide with the one of the Lieb-Liniger model. Here they also did not discuss such a kind of feature.

All in all, I did not recommend this submission for publication in SciPost.

Requested changes

The authors should consider the above questions in their revised version for submission to a suitable journal.

  • validity: ok
  • significance: low
  • originality: ok
  • clarity: ok
  • formatting: reasonable
  • grammar: good

Author:  Wayne Jordan Chetcuti  on 2023-01-27  [id 3278]

(in reply to Report 1 on 2023-01-22)

We thank the Referee for their comments/criticisms. In the following, we provide a point-to-point reply to the comments and questions put forward by the Referee.

R: “ The authors claimed a new method which was built to evaluate the one-particle correlation function of SU(N) fermionic systems. They carried out some calculation of such one-particle density matrix in mesoscopic systems with the particle number less than 10. Although this research topic is of great interest and importance, to my best knowledge, this paper did not reach a satisfactory level of research. Their main result was just a little extension of the method used in Ref.[33], and the previous paper Ref.[42] etc. Here I raise up few questions for their manuscript: ”

  1. R: “The authors claimed their results to be exact. It is true only if their calculations are efficient and correct in terms of Bethe ansatz equations within strong coupling limit. According to their statement, ‘dilute limit’ (Np/L ≪ 1) was necessarily needed. However, in Figs.4 & 6, there are cases that 10 particle on 15 sites. This density is far from the dilute limit, and they did not provide verifications or comparisons with the results based on other methods such as exactly diagonalization (ED) or DMRG. Therefore, it should be of caution with the validity of their results in these Figures. ”

A: We thank the Referee for pointing this out. The Referee is correct in saying that the SU(N) Hubbard model is Bethe ansatz integrable in the dilute limit of Np/L ≪ 1 and that the systems considered in the manuscript are far from this condition. Bethe ansatz integrability hinges on the fact that the scattering of more than two particles does not occur (Yang-Baxter factorization of the scattering matrix). Typically, for SU(N) fermions such a condition can be enforced in two regimes: (i) the dilute limit as stated in the paper; and (ii) the case of Np = L with large repulsive interactions where the model turns into a Sutherland antiferromagnet.

In the infinite repulsive regime, the multiparticle scattering is suppressed since the probability of two particles interacting is vanishing. Therefore, despite the fact that we are far from the dilute limit condition, the system is indeed very close to be integrable. Indeed, both the low lying spectrum and corresponding correlations, numerical diagonalization and Bethe ansatz agree up to 5 decimal places. As it is well know by the community and as pointed out in our manuscript, DMRG is not feasible in the limit of large interactions and large degree of state degeneracies. We agree with the Referee that the manuscript would benefit greatly by highlighting this problem better in the manuscript, in addition to providing some figures in the Appendix to corroborate that our analysis is justified.

  1. R:“They claimed the methods are applicable to SU(N) fermionic system. Obviously, in their main results i.e. Figs.4 & 6, only situations of SU(2) were considered. In fact, for SU(2) fermionic system in strong coupling limit, various methods and results have been obtained in literature, see [33], also see the review articles in this filed, etc. The discussion showed in Fig.5 looks rather rough and lacks rigorousness. The important references for the study of the SU(N) interacting fermions in 1D are missing.”

A: We do not agree with the Referee that our methods and results are restricted solely to SU(2) fermions. Figs. 3 and 5 display results for up to N = 6. The Referee is correct in pointing out that Figs. 4 and 6 correspond only to SU(2) fermions. As discussed in the manuscript and very extensively in the Appendix, due to the fractionalizaiton of the angular momentum reproducing these plots for SU(N > 2) fermions is quite challenging. Nonetheless, we are able to replicate some of the results that we present for SU(N > 2) fermions.

Nonetheless, we want to clarify that we are extending the already exisiting method in [33] by including the flux. This change, although might seem as a small addition, leads to important consequences that have been discussed in [42]. This is confirmed in this work, which actually is shown in the limit of U → ∞ that cannot be addressed numerically.

We do not claim at any point in the manuscript that this is the first instance of calculating correlations in SU(2) fermionic systems in the strong coupling limit. Rather, we expand on the vast literature that exists to take into account the effect of the flux on the system.

The Referee is correct in saying that the discussion for Fig. 5 is lacking.

We would be happy to implement the important references for the study of SU(N) interacting fermions in 1D that the Referee feels that we missed, provided that they point them to us.

  1. R:“The authors claimed their results are well-beyond ED, while the manuscript lacks of explicit proof or com- parison with ED results.”

A: We reiterate that our calculations are well-beyond exact diagonalization because of the size of the Hilbert space. Our method as stated in the conclusion can access a system of 12 particles residing in 38 sites correspond- ing to a Hilbert space of 2 billion, wheras exact diagonalization can only handle around 7 million. Therefore, there is no direct comparison between the two methods. Within the small particle and system size regime, which is reachable by exact diagonalization, we have perfect agreement.

  1. R: “Their results did not show essential physics of SU(N) fermions, such as universal nature of momentum distributions in connection to the Tan’s Contact, as well as to the power law decay of Luttinger liquid, etc. In general, when the component N ≫ 1, the behaviour of ground properties of the SU(N) fermions coincide with the one of the Lieb-Liniger model. Here they also did not discuss such a kind of feature. ”

A: Tan’s contact is related to the TOF. In this paper we have mostly focused on the momentum distribution of the particles, which is related to the momenta the particles occupy. This difference is highlighted by the fact that we specified nT OF . Although interesting, the main focus of the paper is not TOF momentum distribution.

The Referee mentions that we do not make the connection between the ground-state properties of SU(N) fermions and that of the Lieb-Liniger model. Whilst there is no explicit connection to this specific model, in the first paragraph of Section 3 A, we highlight the connection between the momentum distribution of SU(N) fermions and the Bose-distribution. Albeit we agree with the Referee that such a comment is brief and could be expanded further.

---

## Round 1 · Referee Report · Anonymous (Referee 2) · 2023-2-4

Strengths

-the calculation of the one-particle density matrix for the SU(N) Hubbard model (even in limiting cases) is very challenging;

-interesting proprieties of the momentum distribution function on varying the system’s parameters are commented in the manuscript;

-the importance of their results for some experimental measurements is discussed.

Weaknesses

-the exact solution of the model at large U was already discussed in the literature. In this context, it has been only exploited to decouple the charge and spin dof but not to characterize the eigenstates of the model (due to their complicated structure);

-the procedure that is used to find the ground state is rather technical. This may discourage the further application of the proposed method;

-the discussion on the computational advantage of the proposed method w.r.t. other (numerical) methods can be improved (see below);

-the discussion of some parts of the manuscript can be improved e.g. Sec II b and Sec. III A (see below).

Report

In the submitted manuscript, the authors investigate the one-particle density matrix and the momentum distribution function of a SU(N) Hubbard model on a ring in the presence of magnetic flux. Their results for these quantities are based on a Bethe-ansatz solution for the model at strong repulsion that has been previously derived (see refs. [41,42] of the manuscript).

Exploiting such solution to find the ground state of the model (see below), they discuss several aspects of the momentum distribution function, such as the dependence on the number of colors ‘N’ or its behavior with a non-vanishing magnetic flux. The utility of their approach for TOF measurements with ultracold atoms is also discussed.

As the authors comment in Sec.II of the manuscript, a Bethe ansatz solution for the model easily enables the calculation of spectral quantities but it is not very useful for the characterization of the eigenstates (which show a complicated structure even in the limit of large U where the spin and the charge dof are decoupled). Therefore, they propose a method for the calculation of the spin wave function based on exact diagonalization for the spin chain, integrability of Hubbard, and proprieties of SU(N) eigenstates. This is quite an intricate procedure to find the ground state. Although I understand how it is performed and why it is necessary, it is not clear to me to which extent it limits the calculation of the momentum distribution function in terms of N, Np and L, and what is its advantage w.r.t. other numerical methods. This is an important aspect that in my opinion the authors should comment more in the text. Indeed, in the current version of the manuscript, it is only mentioned in the conclusion. Also, in the conclusion, the authors write: “we note that our Bethe ansatz scheme agrees well with numerics of the lattice model, also slightly beyond the dilute regime” but I do not see any direct comparison of their result with numerics in the manuscript.

From the calculation of the ground state, the results for the one-particle density matrix and momentum distribution are obtained with standard methods.

Other questions on the manuscript are collected in the section below.

Concluding, I believe that the manuscript contains interesting results for the community and it deserves to be published after a revision according to the points below. However, in my opinion, the results proposed by the authors are not enough groundbreaking to fully meet the acceptance criteria set by SciPost Physics. For this reason, I recommend the publication of the revised manuscript in SciPost Physics Core.

Requested changes

-Page 3 eq.(5) – I can understand what the authors mean with this writing and Fig.1 further explains the spin-charge decoupling occurring at large U but the notation can be improved. Indeed, in its current form, it is hard for me to give a precise meaning to Eq.(5)

-Page 3 section II (a) – the authors should comment more the advantage of their method to find the ground state of the SU(N) Hubbard model. It is not clear to me what computational limit one should expect in terms of color number N, particles Np and system size L. In addition, it would be very helpful to show the comparison of their method with numerics for the lattice model, as mentioned by the authors in the conclusion.

-Page 3 section II (b) – the discussion on the one-particle density matrix can be improved. The relation between lattice operators and the continuous fields $\Psi_\alpha$ can be written down explicitly. Also, I imagine that the continuous description is needed to exploit the integrability of the model for N>2, as discussed in the introduction. However, this is not commented in the text by the authors making this part not very clear for unexpert readers.

- Page 4: the authors write “The spin-charge decoupling is obtained by Bethe equations in addition to exact diagonalization of the Heisenberg model and DMRG”. I do not understand what is the precise meaning of this statement by the authors.

- Page 5, section III A – I find this section rather unclear although I can make an effort to understand the authors’ results (at least up to a certain extent). In particular, 1) the limit of large U is not mentioned at all in the text and in the figure 5 caption (which I assume concerns this specific limit). 2) The quantity $\Delta$ (which I assume is the Fermi gap) has not been introduced. 3) The Fermi gap is defined as $f(k_f)-f(k_f+\Delta k)$ but I do not see any definition of $f(k)$ (which I assume being the Fermi-distribution).

- Figure 5 – Do the authors have an intuitive understanding of the non-monotone behavior of the Fermi gap at N=3?

---

## Round 1 · Referee Report · Anonymous (Referee 3) · 2023-2-9

Strengths

1) SU(N) fermions is a very hot-topic both for theoreticians and experimentalists
2) The method used allows to deal with a "quite large"number of particles and components

Weaknesses

1) The article is not adequately well written (description of the method)
2)The methodology is not new
3) The physical discussions are lacking

Report

The authors of this work have studied a gas of repulsive N-components fermions confined in a ring-shaped potential, subject to an effective magnetic field, in the limit of large repulsion strengths.
They split the problem into the spinless fermionic and SU(N) Heisenberg models, in order to succeed in calculating the one-body density matrix and thus the momentum distribution up to 10 particles and 6 components (N_p=10 with N=2 and N_p=6 and N=6).
The authors claim that their achievements are well beyond the current state-of-the-art tractable by numerical methods.

My main critcisms are the following:

1) the method, as the authors say, is not new. This is an extension to the case with a flux. So this work could be the opportunity to
- give some more insights on the method
and
- to discuss some physical aspects of such a system

In my opinion none of these two points has been developped adequatly.
Even if they show some TOF interference patterns, the physical discussion is missing. Morover they authors could study the beviour of the k=0 mode and of the tails of the momentum distribution.

2) Concerning the current state-of-art for DMRG calculation applied to SU(N) fermions, they authors should read in details Ref. [52] (N_p max=12 and N max=6)

I invite the authors to revise deeply their manuscript. I think that their work may deserve to be published in SciPost, but not in the actual form.

Requested changes

I invite the authors to revise deeply their manuscript, first of all introducing more physical discussions.

Moreover I suggest them to revise the technical description. Here below they will find some, not exhaustive, explicit points:

1) In the introduction the authors have written: "In the dilute regime of few particles per site, the lattice model captures the physics of continuous systems.."
This has to be changed in: "In the dilute regime of less than one particle per
site, the lattice model captures the physics of continuous systems.."
Indeed, as the same authors have written at the end of the first column at page 2, $\nu\ll 1$ (when considering the dilute regime)

2) Below Eq. (2), the authors refer to the rapidities \Lambda_\beta: these rapidities are not introduced anywhere and do not appear anywhere either.
I invite the authors to introduce explicitly the formalism, including all the needed definitions. I know that they are in the appendices, but the paper has to be "self-consistent" for a reader even if some details are specified in the appendices.

3) same type of comment concerning the "the SU(N) quadratic Casimir operators" (middle of column one at page 3): they are defined in the appendix, but here the authors may include some physical insights

4) Which is the normalization of the momentum distributions in Fig. 3? It is not N_p. Is it the number if fermions per component? Please, specify in the caption for the main figure and the inset.

5) There is a problem in the caption of Fig. 4 for the definition of n_{+} and n_{-} (actually n_{-} is identically zero!)

6) the term ω(j → l, α) is introduced at the end of the first column at page 4, but is explicitly defined only in Eq (9) (after 14 lines!).

7)... etc...

8) please, revise your comments concerning DMRG calculations

---

## Round 2 · Referee Report · Anonymous (Referee 2) · 2023-3-30

Report

In the revised version of the manuscript, the authors correctly implemented the points mentioned in my previous report (in particular Appendix D where they discuss the comparison with other numerical methods).

Therefore, I recommend the publication in SciPost Physics Core.

---

## Round 2 · Referee Report · Anonymous (Referee 1) · 2023-3-31

Strengths

The authors presented a method on the study of the one-particle correlation function in the SU(N) sermonic Hubbard model. This is a challenge problem in theory. Although the results look unlike what I expected, the quality of the submission is still fine and reach the bottom line of the Scipost standards.

Weaknesses

Their method is general and model dependent.

Report

I would like to recommend this submission to publish in SciPost.

Requested changes

No more change is requested.

---

## Round 2 · Referee Report · Anonymous (Referee 3) · 2023-4-11

Strengths

1) studies cocerning SU(N) fermionic systems are of great interest form a theoretical and experimental point of view. 2) The limit of large repulsion strengths is difficult to be treated numerically

Weaknesses

1) The method presented here is not new

Report

The authors have answered to all my criticisms and suggestions. I support the publication of this manuscript in SciPost.

Requested changes

I have no further requests

---

## Round 2 · Author Response

Reply to the Referee reports is included alongside the re-submitted manuscript, where the changes are highlighted in blue.

---

## Editorial Decision

published